# IoT-based Smart Water Level Monitoring

Ritik Yelekar, Thomas David Tency, Nagesh Walchatwar, Sahil Padole

**Abstract**

This report aims to address the limitations of current water level monitoring systems, which are bulky, expensive, and difficult to maintain, resulting in limited deployment. To overcome these challenges, the study utilises the Internet of Things (IoT) - enabled low-cost sensor nodes for water level monitoring. In this study, an ultrasonic sensor-based water level node is developed to send data to the cloud through GPRS (2G). Five such nodes were deployed to monitor water levels in overhead tanks and sumps on the campus of IIIT Hyderabad, India. The collected water level data were analysed for behavioural patterns and detecting faulty float switches. In addition, a deep learning algorithm was employed, which can predict future water needs. Thus, the proposed IoT-based smart water level meter offers a more accessible and cost-effective approach to water level monitoring.

## 1  Introduction

Water is a fundamental necessity for human life. Water monitoring is crucial for effectively managing this precious resource as the demand for water rises. Continuously tracking water levels empowers authorities to make informed decisions regarding water allocation, conservation measures, and distribution optimisation, especially in water-scarce regions and urban areas with surging water demands. The Internet of Things (IoT) has revolutionised water level monitoring, introducing remote, real-time, and continuous measurement capabilities. IoT-based systems offer potent data analytical capabilities, processing big datasets to identify trends, patterns, and anomalies in water level behaviour. This project focuses on IoT-based water level monitoring in overhead tanks and sumps.

## 2  IoT Implementation and Field Measurements

### 2.1  Sensor Node Implementation

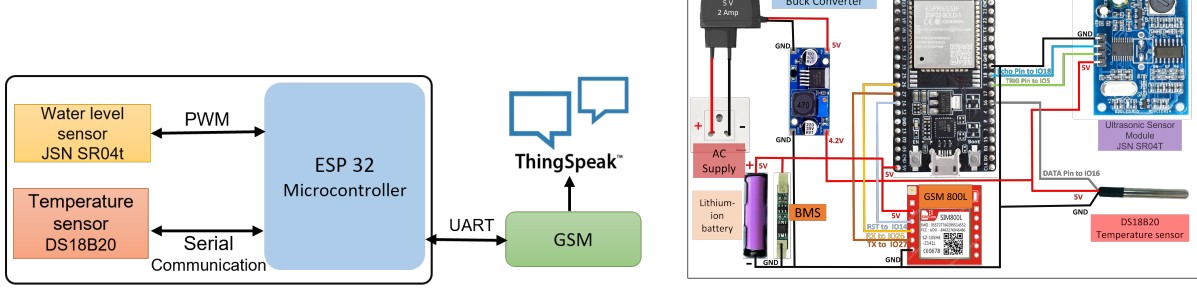

Figure 1: Block Architecture and Circuit Diagram

Table 1: Specification of sensors used in the deployed sensor node

| Sensor | Parameters | Resolution | Accuracy |
|---|---|---|---|
| JSN-SR04T | Distance | 1mm | +-1cm |
| DS18B20 | Temperature | $0.1^{\circ}$C | +-$0.5^{\circ}$C |

Figs. 1 show the block architecture and circuit diagram of the smart water level monitoring sensor node developed and deployed in the IIIT-H campus. Each deployed node consists of ESP32 [1] as a main microcontroller, water level, and temperature sensor. Table 1 shows the specification of sensors used in the deployed sensor node. JSN SR04T [2] is the ultrasonic sensor used as a water level sensor in this hardware. It is waterproof, has a more extended range, and has higher accuracy than sensors such as ultrasonic HC- SR04 [3]. The temperature sensor used is the DS18B20 [4], which offers a digital output, a wide temperature range, low power consumption, and a waterproof package, making it ideal for in-tank temperature measurement. The microcontroller processes data from both sensors and sends it to ThingSpeak [5]. This cloud-based IoT platform allows users to collect, analyse, and act on data from sensors or devices. The data collected from the ultrasonic and temperature sensors can be analysed and displayed in real-time, making monitoring and managing the tank water level easier. The data is sent using GSM800L [6] for monitoring and to alert the user when the water level is too high or too low or when there is a problem with the system. The LM2596 DC-DC buck converter [7] regulates voltage levels in the system and ensures that all components receive a consistent power supply. The lithium-ion battery of 1500 mAh is a backup power source in case of power outages. Battery Management System (BMS) monitors and controls the charging and discharging of the lithium-ion battery and protects it from overcharging, discharging, and short circuits. Fig. 2 shows a deployment-ready sensor node, which consists of sensors, a microcontroller, and all other components, which are all enclosed in a poly-carbonate box of IP65 rating as the deployment is outdoors. IP65 enclosures offer complete protection against dust particles and a good level of protection against water. These nodes are designed to be placed on top of a tank.

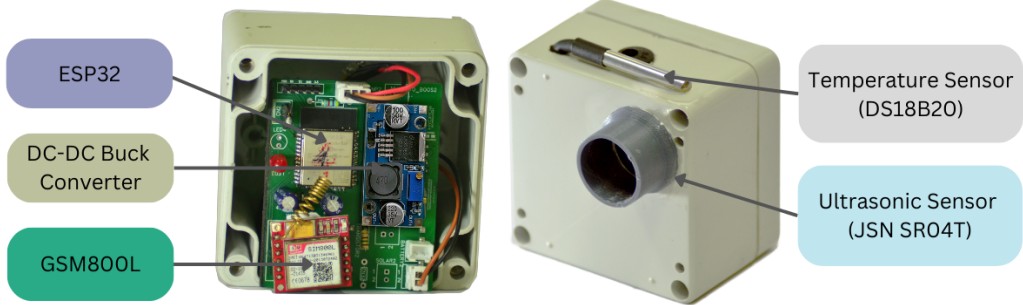

Figure 2: Water level monitoring node

## 2.2    Proposed Methodology

This section will delve into the fundamental working principles of the ultrasonic sensor as a water level measuring device and explore the purpose behind incorporating a temperature sensor within our IoT-enabled sensor node. The weather-proof design considerations that ensure the sensor node's reliability and functionality in various environmental conditions will be discussed. The ultrasonic sensor employs the Pulse Width Modulation (PWM) communication technique. This device operates at a frequency of 40 kHz. Ultrasonic waves can travel through the air with little attenuation, making them well-suited for distance measurements. We are using the Time of Flight (TOF), which is the time period a sound wave takes to travel from the ultrasonic sensor to the water surface and back to the sensor. In this case, the time of flight is used to calculate the distance between the sensor and the water surface. The temperature sensor measures the temperature of the air within the tank, which helps the system compensate for changes in the speed of sound due to temperature fluctuations, resulting in high system accuracy.

The relationship between distance of sensor from water surface $(L)$, time of flight $(t_f)$, and the speed of sound $(V_s)$ can be expressed using the formula [8]:

$$L = \frac{t_f \cdot V_s}{2},\qquad(1)$$

taking $V_s$, which is 343m/s as the default value of the speed of sound at room temperature, would result in an error in the water level calculation.

## 2.3    Theory used in temperature compensation

The temperature compensation technique was used to incorporate the effect of air temperature in the tank for calculating the speed of sound. The speed of sound, incorporating the temperature dependence [8] can be given as

$$V_{wtc} = 343\sqrt{\frac{T + 273}{273}}, \tag{2}$$

where $V_{wtc}$ is the dynamic velocity of the sound with respect to temperature, and $T$ is the air temperature in unit °C. The $V_s$ in (1) is replaced with the value of $V_{wtc}$ calculated above (2), giving us $L_{wtc}$, distance with respect to temperature compensation, which can be given as

$$L_{wtc} = \frac{t_f \cdot V_{wtc}}{2}. \tag{3}$$

## 2.4    IoT Network Deployment Strategy

Fig. 3 shows the deployed water level monitoring nodes at the IIIT-H campus. The nodes were deployed across an area of approximately 66 acres ($0.267\ \mathrm{km}^2$) to study the variability of water levels in different environments, buildings, populations, and regions. Node-1, Node-2 and Node-3 were deployed on Over Head Tanks (OHT), whereas Node-4 and Node-5 were deployed on the water sumps known as Pump houses. Table 2 shows the dimensions of each of the tanks and the following points show the buildings in which the nodes have been deployed.

The campus was divided into two categories based on the environment and population patterns:

- *Residential buildings*: Three nodes (Node-1, Node-4 and Node-5) were installed in residential areas. Node-1 was deployed in Parijat block C. This was due to the large number of students residing in this area, allowing us to observe the relationship between the number of people and water consumption levels. In contrast, Node-5 and Node-4 were deployed in two of four main water sumps, Pump House 3 and Pump House 4, which supply water to faculty quarters and Bakul hostel, respectively.

- *Administrative buildings*: Two nodes (Node-2 and Node-3) were installed in administrative areas. Node-2 and Node-3 were deployed in the Kohli research block and Vindhya block respectively. This was because the occupants of these buildings mainly utilise water during office hours and lab hours.

This strategic placement of nodes allows us to determine the different behavioural patterns and identify opportunities to conserve water resources.

Table 3 shows the number of data points available per node. Each of the five nodes independently collects data, namely water level and temperature readings, every 1.5 minutes (90 sec). To transmit this data to a centralised location for further analysis and monitoring, each node utilises the ThingSpeak platform. The server parses the parameters provided in the request and stores the data in the corresponding channels associated with each node. This stored data can be analyzed in real-time, making remote monitoring of water levels possible.

Table 2: Tank dimensions

| Nodes | Length (m) | Breadth (m) | Depth (m) | Tank Volume ($m^3$) |
|---|---|---|---|---|
| Node-1 (OHT) | 9 | 4.5 | 1.2 | 48.6 |
| Node-2 (OHT) | 13.16 | 8.75 | 1.2 | 138.1 |
| Node-3 (OHT) | 6.3 | 2.2 | 1.2 | 16.6 |
| Node-4 (Sump) | 15.2 | 5.27 | 4.4 | 352.4 |
| Node-5 (Sump) | 5 | 2.97 | 2.5 | 37.1 |

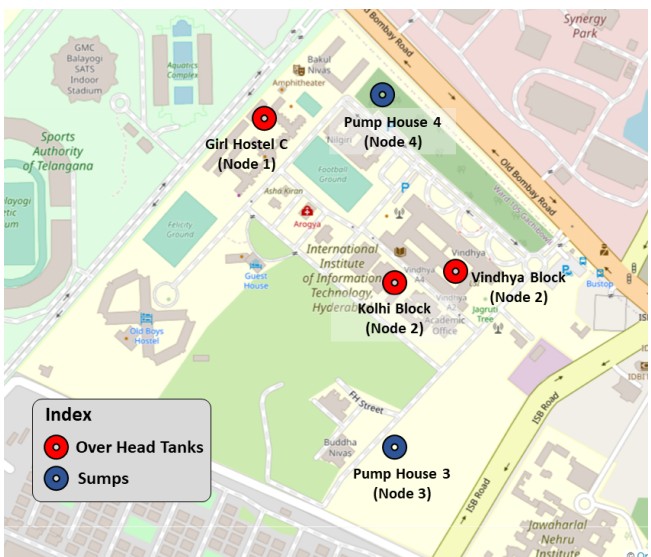

Figure 3: Deployment

Table 3: Number of samples in each Node

| Nodes | No. of data points |
|---|---|
| Node-1 (OHT) | 15801 |
| Node-2 (OHT) | 15982 |
| Node-3 (OHT) | 14104 |
| Node-4 (SUMP) | 16131 |
| Node-5 (SUMP) | 16010 |

## 3   Data Processing Methods

After the data is sent to the ThingSpeak platform, it is essential to process the data to ensure it is in a suitable format for analysis. The following steps are employed to convert the obtained raw data from the sensors into a proper dataset:

### 3.1   Outlier Removal

It is essential to remove the outliers in a raw dataset as there are few extreme values that deviate from other samples in the data, which might be a result of several factors such as:

- Formations of spider cobwebs on the sensors after the node is deployed.

- Changes in humidity in the atmosphere can affect the accuracy of the readings obtained since the moisture in the air can scatter or absorb ultrasonic waves.

Primarily, values exceeding the maximum and minimum values of the true data were removed. A moving window in the mean mode of operation with a window of 5 was run over the entire data to remove any sudden spikes in the data. This gave a smooth trail of data points void of sudden troughs and crests.

### 3.2   Conversion from level to volume

The water level data ($W_L$) can be calculated as difference between depth ($d$) and sensor reading with temperature compensation ($L_{wtc}$)

$$w = d - L_{wtc} \tag{4}$$

It is then multiplied by the length ($l$) and breadth ($b$) of the tank, giving us the volume ($V$) of water in the tank.

$$V = w \cdot l \cdot b \tag{5}$$

These volume values are used for further analysis.

## 3.3    Moving Average

The data has been averaged using a simple moving average method to remove unnecessary fluctuations in the data that generally happen in water tanks due to the slight water movement. The data has been averaged to one sample per 15 minutes.

## 3.4    Fill rate and consumption rate calculation

For calculating these parameters, a segment of data points is considered where the overall trend is either positive or negative. Linear regression is then performed to find the best-fit line for that segment. If the slope of the line is positive, it is considered fill rate while if it is negative, it is considered consumption rate. Note that these are average values over time as many places simultaneous fill and consumption may happen.

## 3.5    Prediction of future water consumption

Prediction of water consumption can help communities and regions adapt to changing water availability and plan for potential water shortages or surpluses. It is also essential for optimizing water usage, ensuring consistent supply, and avoiding wastage. In this paper, Long Short-Term Memory (LSTM) neural networks have been employed to predict future water demands.

The architecture's ability to capture temporal patterns and correlations in data is particularly advantageous for forecasting water consumption in overhead tanks, where past water usage significantly influences future consumption.

The dataset is preprocessed to handle missing values, outliers, and any inconsistencies that may affect model performance. The data values are then normalised using the Min-Max scaling[9] where the input features are scaled to a range between 0 and 1. This is performed to feed the dataset into Deep-Learning models for further analysis and predictions. Time series data is structured into sequences, and a sliding window approach is applied to create input-output pairs for training the LSTM model. The entire dataset was split into training and testing sets in an 80-20 split. The training set is to train the data, and the testing set is to evaluate the overall performance of the model.

The Keras [10] library in Python was used to implement the LSTM model. The model consists of 3 input layers each consisting of 128 units, 64 units and 32 units respectively. It is followed by a dense layer with a single output. The learning rate was set to 0.01 along with early stopping to avoid overfitting of the data. The Adam optimizer was used to calculate the individual adaptive learning rates of each parameter. To assess the predictive accuracy of the LSTM model, Root Mean Squared Error (RMSE) is employed.

## 3.6    Seepage Detection

Seepage detection within residential water storage tanks is a meticulous procedure involving the systematic identification and resolution of issues such as leakage, structural cracks, and seepage occurrences. These tanks play a fundamental role in residents' daily lives, serving as essential reservoirs for various purposes, including drinking, cooking, bathing, and irrigation, as is the case in our IIIT-Hyderabad campus.

In our specific context, where water conservation is paramount due to the residential setting, the process of seepage detection extends beyond routine maintenance. It assumes a crucial role in addressing water wastage, thereby contributing to our commitment to environmental responsibility. Timely identification and rectification of seepage not only reduce water loss but also support the overall sustainability of our residential community. Additionally, this vigilant approach helps safeguard the integrity of the tank and its surroundings, ensuring the continued reliability of our water supply system.

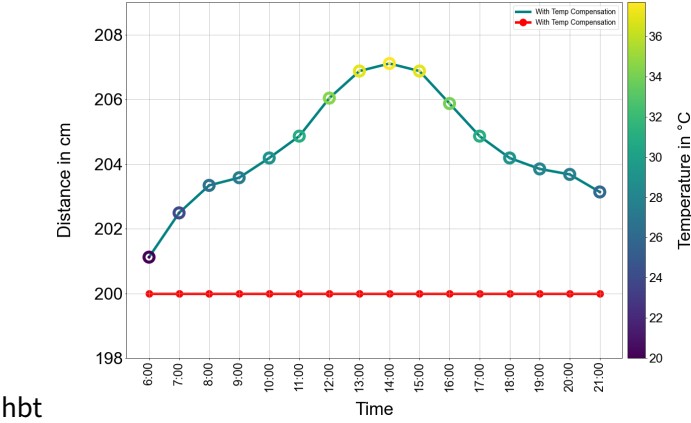

hbt

Figure 4: Results of temperature compensation

# 4 RESULTS AND ANALYSIS

## 4.1 Temperature Compensation

In an experiment to prove the working of temperature compensation, the device was kept at the height of $200$ cm (true distance) facing the ground. In theory, the distance data collected should remain the same, but as the experiment progressed, it was found that the temperature and distance calculated were directly proportional. A maximum deviation of 7 cm was observed when the temperature reached 37.7 °C. On performing temperature compensation over the same data, the distance values reached significantly near to the true value(200 cm) with a precision improvement from 96.5 % to 99.7 %. Hence, temperature compensation corrected the deviation in the water level readings which could be subjected to change with temperature.

## 4.2 Time Series Data

Table 4: Average consumption, Upper Threshold volume and Baseline volume of each tank

| Nodes | Average Consumption (kL/day) | Upper Threshold Volume (kL) | Baseline Tank Volume (kL) | Tank Volume (kL) |
|---|---|---|---|---|
| Node-1 | 10.56 | 36.45 | 14.6 | 48.6 |
| Node-2 | 19.85 | 96.6 | 57 | 138.1 |
| Node-3 | 1.20 | 9.96 | 4.15 | 16.6 |
| Node-4 | 24.42 | 317.6 | 153.19 | 352.4 |
| Node-5 | 2.17 | 29.68 | 10.39 | 37.1 |

Fig. 5 shows the volume of water in five different tanks between 13 January 2023 and 27 January 2023. The plots suggest that each tank's filling and consumption pattern is unique. Table 4 shows each tank's average consumption, baseline volume and upper threshold volume.

A few observations are as follows:

- *Node-1:* Located in block C of the Parijat hostel which houses a tank with a capacity of 48.6 kL (48.6 $m^3$). It fills at a rate of 1.296 kL/h and consumes water at a rate of 0.64 kL/h. It takes around 13 hours to fill the tank to the desired level and approximately 21 hours for it to be completely depleted. To ensure efficient water supply, the tank's baseline level is 30% of its total capacity, equivalent to 14.6 kL.

- *Node-2:* Located in the Kohli Research block, it houses a significantly larger tank with a substantial volume of 138.1 kL (138.1 $m^3$). This tank serves a dual purpose, supplying water to the block and functioning as a reservoir for the fire-retardant network. Operating with a fill rate of 5.52 kL/h and a consumption rate of 1.53 kL/h, it takes approximately 6 hours to reach the desired fill level and around

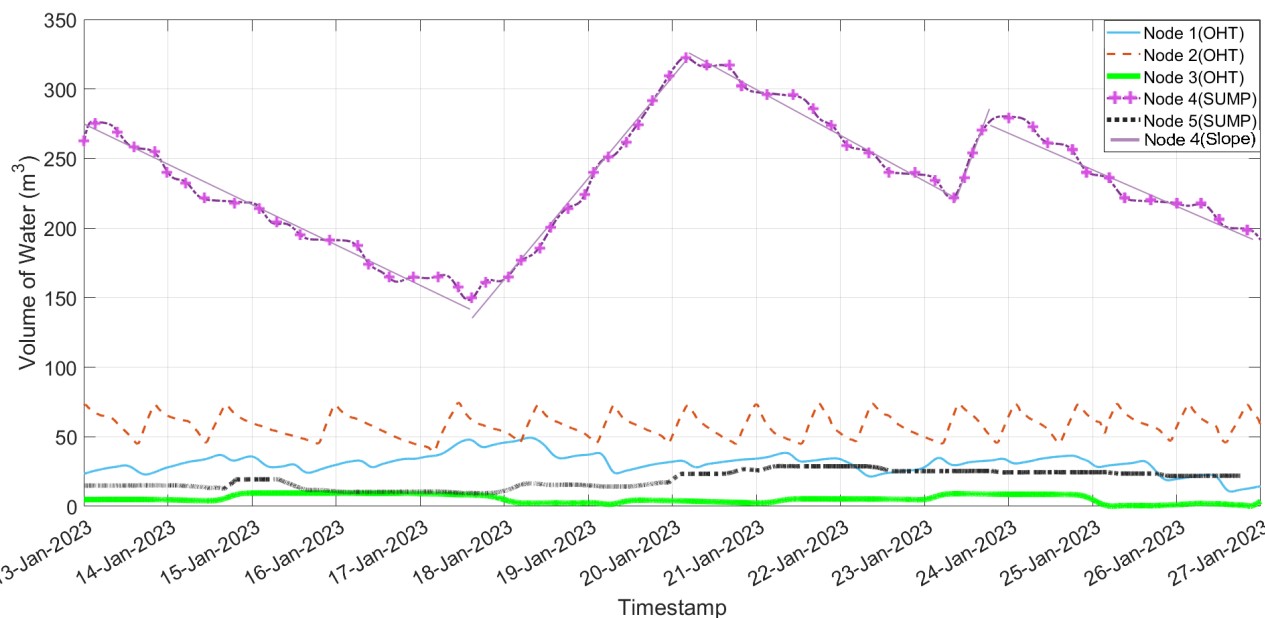

Figure 5: Volume of water vs time for all the five tanks

18 hours to deplete its contents entirely. The tank's baseline water level is meticulously set at 40% of its total capacity to ensure efficient water supply to the block, amounting to 57 kL.

- *Node-3:* Located in block Vindhya, it has an automated tank to store drinking water. Despite its relatively small volume of 16.6 kL (16.6 $m^3$), the tank is vital in ensuring a reliable water supply. The fill rate of the tank is 0.32 kL/h, while the consumption rate is 0.25 kL/h. It takes approximately 5 hours to fill the tank to the desired level and around 8 hours to deplete its contents completely. To ensure an efficient water supply to the block, the tank's baseline water level is 25% of its total capacity, corresponding to approximately 4.15 kL.

- *Node-4:* Located in Pump House 4 (PH4), a non-automated and underground water sump collects water from a borewell and other sumps. The tank at this location boasts a significant capacity of 352.4 kL (352.4 $m^3$). The tank's fill rate is 3.36 kL/h, while its consumption rate is 0.919 kL/h. The tank adheres to a predetermined schedule that aligns with technician shifts to ensure a consistent water supply. Over 2.6 days, which spans 62 hours, the tank undergoes six cycles of motor operation to achieve complete filling. Each cycle consists of 7 hours dedicated to filling, followed by a resting period during which the motor is switched off. On the consumption side, it takes approximately 4 days to consume the tank's contents entirely. For efficient water supply management, the tank maintains a baseline water level of 40% of its total capacity, approximately 153.19 kL. An upper threshold has also been established at 90% of the tank's volume, equivalent to around 317.16 kL. The tank is filled at different intervals based on water and technician availability.

- *Node-5:* Located in Pump House 3 (PH3) and is a non-automated water sump. PH3 is an underground water tank collecting potable water from the Manjeera water supply. Despite its relatively small size, with a storage capacity of 37.10 kL (37.1 $m^3$), it efficiently meets the water needs of the faculty quarters. The tank has a fill rate of 0.90 kL/h and a consumption rate of 0.30 kL/h. It takes approximately 3 hours to fill the tank to the desired level and 9 hours to deplete its contents. The tank maintains a baseline water level of 30% of its total capacity to manage the water supply, around 10.39 kL effectively. An upper threshold has been established at 80% of the tank's volume, corresponding to roughly 29.68 kL. The water level data reveals inconsistencies in the sump filling process, as a technician manually operates the pumps when the water level is low.

By studying this consumption and filling patterns, valuable insights were gained. We could identify trends specific to each tank, which helped us understand how efficiently water was being used and how the float switch mechanism affected water levels.

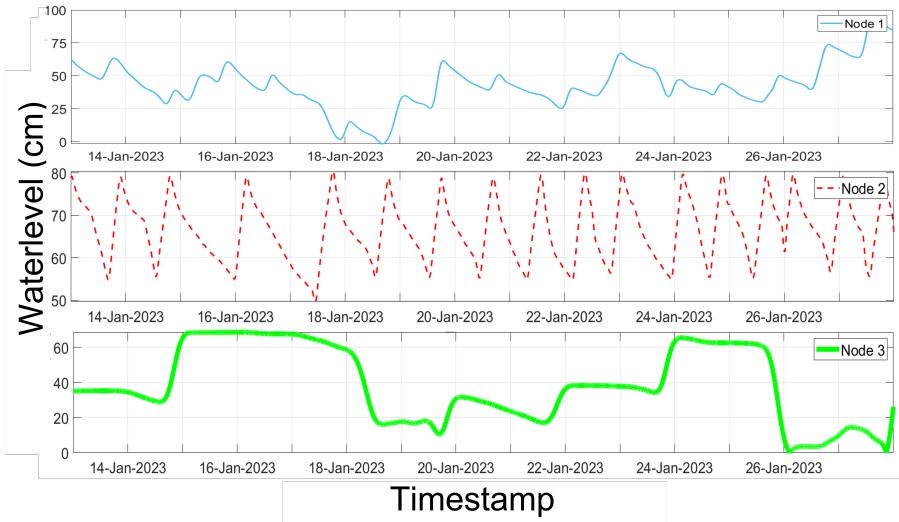

Figure 6: Time series water level plots of Nodes 1, 2, and 3 for the OHT Tanks.

### 4.3    Faulty Float Switch Detection

The management of IIIT-H has installed float switches in almost all OHTs to control the activation and deactivation of the water supply motor. Fig. 6 shows that some of the tanks are only filled up to approximately half of their total capacity. This configuration was implemented to prevent the tanks from being filled to their maximum capacity, allowing for complete daily water utilisation. The intention was to avoid large amounts of dead storage, which could compromise the water quality within the tanks.

Note that these float switches ensure that the motor operates when the tank is empty and shuts off when it reaches its total capacity. This results in a nice pattern where the motor turns ON when the float switch reaches around the set low point, and the motor switches OFF when the float switch reaches around the set maximum point. This pattern can be seen clearly from Fig. 6 for Node-2, for which the float switch is working fine. However, it can also be seen from the figure that the float switches in OHTs corresponding to Nodes-1 and 3 are not following the pattern and the motor is switching ON and OFF slightly away from the set points, indicating that the float switches are faulty. This observation was also confirmed by the management. Thus, based on the water level data, faulty float switches can be easily detected by monitoring the deviation in the starting and stopping points of the motor as compared to the set points.

### 4.4    Prediction of water consumption

Figs. 8 and 9 show the results of prediction of water consumption using LSTM for the Nodes-1, 2, and 3 corresponding to the overhead tanks. Node that first 80 % of the data is used for training while the last 20% the data is used for testing. The black curves indicate the predicted consumption. It can be seen that the LSTM is working very well and the error in the prediction is very small, which can be confirmed from Table 5, which shows the RMSE value for the predicted curves.

Table 5: RMSE of OHT nodes

| Nodes Name | RMSE (in cm) |
|---|---|
| Node-1 | 1.56 |
| Node-2 | 2.40 |
| Node-3 | 1.57 |

### 4.5    Seepage Detection

On processing the data of the deployed devices. Pump House 4 pushes water into and out of the sump at intervals, acting as a reservoir to pump water to Bakul Boys Hostel. This means that water is not being

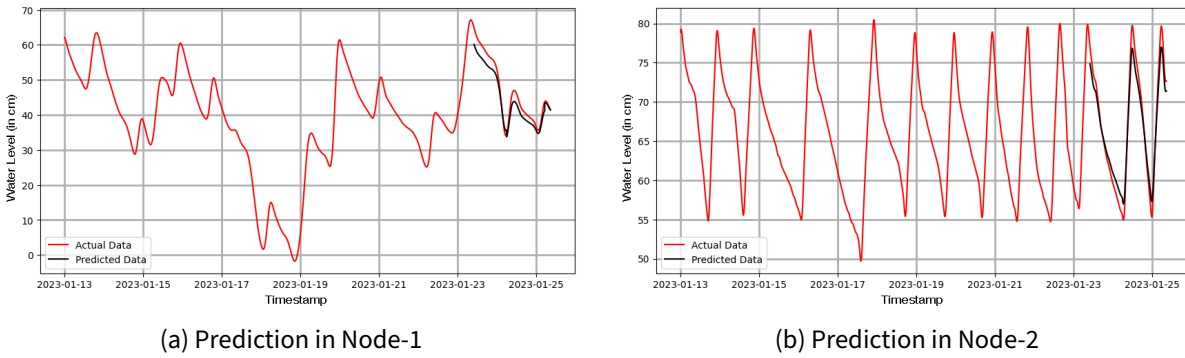

(a) Prediction in Node-1          (b) Prediction in Node-2

Figure 7: Predictions in Nodes 1 and 2

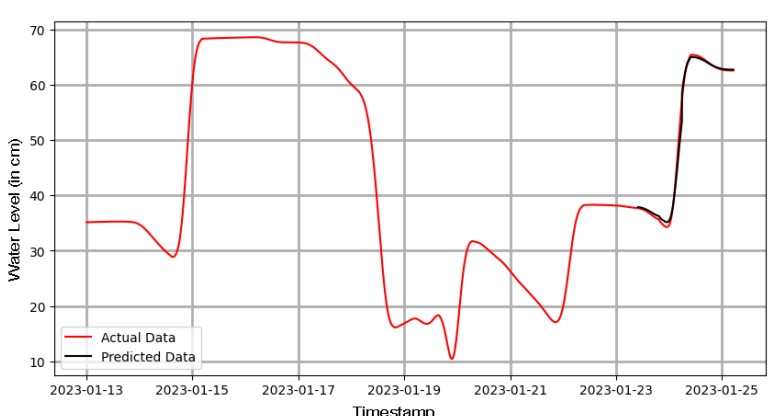

Figure 8: Prediction in Node-3

constantly used or being pumped in this sump in modes of live consumption such as drinking, washrooms, etc. Here, water has a fixed inflow and outflow rate. On observation of the data from 30th August when the pump house motors were shut down to make sure no water is pumped into and out of the system, it was observed that the water levels of the tank decreased by 1cm over the 12-hour gap. The dimensions of tanks being $15.4\,\mathrm{m} * 5.2\,\mathrm{m}$, equates to the total water lost being equivalent to

$$\mathrm{Vol}_{\mathrm{WL}} = \mathrm{L}^* \, \mathrm{B}^* \mathrm{D}$$

Where Vol $\mathrm{WL}_{\mathrm{L}}$ is the total volume of water lost $L$ is the length of the tank $B$ is the breadth of the tank $D$ is the difference between the water level when the pumping was shut off.

$$\mathrm{Vol}_{\mathrm{WL}} = 15.4 * 5.2 * 0.01$$

$$\mathrm{Vol}_{\mathrm{WL}} = 800.8 \; l$$

A substantial water loss of around 800 litres was observed due to leakage on 30th August 2023. This detection was only possible after placing such devices in sumps to monitor water levels of the same. We notified the IIIT-H water management team about this problem. Detection of such leakage with such devices can help address the problem faster and help save a lot of water.

## 5   Conclusion

The findings in this study demonstrate how low-cost IoT-enabled water level monitoring nodes can be used to monitor the water level in both overhead tanks and underground sumps, thereby improving the efficiency of the use of water resources. Using low-cost waterproof ultrasonic sensors made the nodes more econom­ical and robust. The use of temperature compensation proved to improve the precision of detecting water

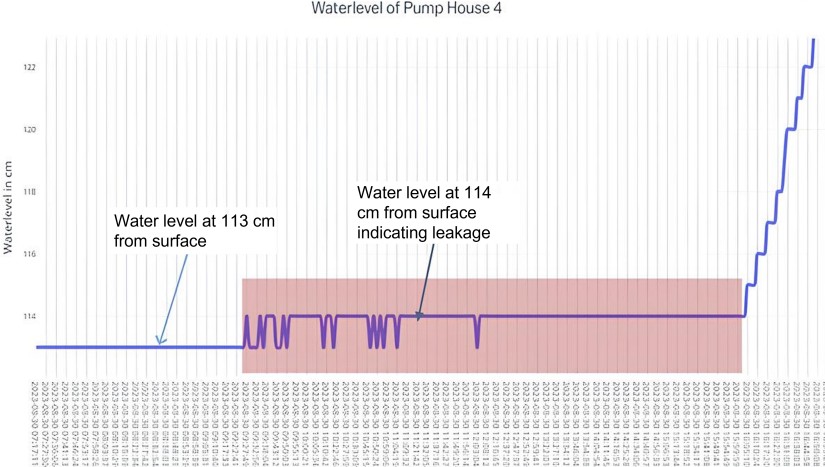

Figure 9: Seepage Detection

levels from 96.5% to 99.7%. The five smart devices were deployed in different locations on the IIIT-H campus. The data collected from these devices provide significant insights into the water infrastructure of the campus. The study introduced float switch fault detection, which assesses the working status of the device responsible for switching motors to avoid overflow and empty states. Considering the chance of the device getting shut down and for predicting future usage, this study also predicts water level using LSTM with a mean RMSE of 1.8 cm. This IoT-based water level monitoring system offers a promising low-cost solution for monitoring water distribution and minimising water wastage with seepage detection.

## 6 Availability

1. Source Code

2. Demonstration Video

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
