# OpenReview forum: "IoT-based Smart Water Level Monitoring"
_helsinki.fi/ESPC/2023/Competition — ESPC 2023 LongPresentation_

### Official Review · Reviewer_tjR9 · 2023-11-13

**Rating:** 3
**Confidence:** 3

**Summary:**

The authors present a system that uses an ultrasonic sensor for measuring the water level. Their solution has been deployed at 4 locations in their campus, and their deployment has aided the monitoring the consumption and also identifying leakages.

**Strengths:**

- The system includes a temperature sensor to account for impact of temperature on the speed of sound.
- The authors have provided insights on the factors that cause outliers in practical deployments.
- The system can be used for detecting leaks which is vital for conserving water.
- The system has been deployed on tanks that are being actively used.

**Weaknesses:**

- The reasons for the interval used for computing the moving average have not been provided. Hows does changing the moving average interval affect the measurements?
- It is not clear why the predictions were not performed for nodes 4 and 5.

Minor nitpicks:
- The video blanks out at the 7 minute 30 seconds intervals.
- The code documentation can be improved. It contains only one text file without any instructions on how to use the file.
- Some insights on the message formats used when pushing data to thinkspeak would have been useful.
- The conversion from level to volume implies that the shape of the tank is box shaped and the report can implicitly given animpression that would work with only box shaped tanks.
- Figures 7 a , 7 b and 9 are difficult to read and they require a significant zoom to read the axis labels.

---

### Official Review · Reviewer_Hbpe · 2023-11-17

**Rating:** 3
**Confidence:** 2

**Summary:**

Article proposes IoT solution for water level monitoring. The system is fully designed, and implemented. IoT nodes are deployed to different sites for short testing.
The authors also provide some exploratory data analysis, as well as LSTM model for water consumption prediction.

**Strengths:**

- System is fully functional
- Explorative analysis is performed with some findings regarding water consumption behaviour
- LSTM model presented for water consumption prediction
- Report is generally well written

**Weaknesses:**

Overall, nice work to be presented at the competition. Maybe little more details could be done regarding particular selection of the machine learning model and why certain architecture decisions made? Some references to related work would enhance the report as well.

---

### Official Review · Reviewer_VLqZ · 2023-11-17

**Rating:** 2
**Confidence:** 3

**Summary:**

This report presents a low cost IoT based system for water level monitoring.

The node is developed with an ESP32, temperature sensor, ultrasonic sensor, GSM chip, and DC-DC buck converter.
The node is waterproof and collected data is sent to ThingSpeak cloud server for further processing.
The prototype is deployed at five different sites in the authors' university campus.
Main principle behind measure water level is the time of flight for sound waves - the time taken by sound waves to travel from the node to the water and come back.
Their prototype measures the volume of water and also predicts future water consumption.

**Strengths:**

*The prototype works in an untethered mode i.e. no strings or wires needed.
*Data is being sent to cloud for further processing.
*Future water needs can be predicted
*Analysis done for various sites
*Module for seepage detection is also implemented

**Weaknesses:**

*Many mistakes across the report. Here is an example: Each deployed node consists of ESP32 [1] as a main microcontroller, \textcolor{red}{water level}, and temperature sensor.
*Figure 4 has a stale hbt written
*No ground truth given. How is the accuracy of the system established?
*Details on how seepage detection is working are missing
*Logic behind water prediction is not clearly explained

---

### Official Review · Reviewer_NKLk · 2023-11-18

**Rating:** 3
**Confidence:** 4

**Summary:**

This project provides an IoT-based solution to monitor and calculate the water level (filled-in water tanks). The project uses an ultrasonic sensor-based water level node to calculate the water level and uses GSM to transmit data to the Thing Speak cloud. The project then deploys five of the developed sensor systems at IIIT Hyderabad to collect data. Using the data, the project carries out a data analytics and error analysis and then applies LSTM to forecast water consumption.

**Strengths:**

The project presents a cost-effective solution for water tanks. The project considers the water level measurements, seepage detection, water consumption prediction (using LSTM), and faulty switch detection. The use of ultrasonic sensors to calculate the water level (through distance) is also a nice approach.

**Weaknesses:**

The novelty of the work is still unclear. The report does not compare the solution with the existing other methods. The report does not explain how the data is transmitted in detail. For example, how the GSM works and what is the frequency of data transmission. The size of the dataset also has not been mentioned in the report.

---

### Official Review · Reviewer_WAMi · 2023-11-20

**Rating:** 4
**Confidence:** 3

**Summary:**

The study utilised low-cost sensor nodes for water level monitoring. A ultrasonic sensor-based water level node is developed to send data to the cloud. Five such nodes were deployed to monitor water levels in overhead tanks and sumps on the campus of IIIT Hyderabad. The water level data were analysed for behavioural patterns and detecting faulty float switches and a deep learning algorithm was employed, which can pre-dict future water needs. The report is well written and informative. The results and finding are interesting. The use of ML to make predictions is good. However, the report missing a  good introduction leading to a research question. The measured storage tanks and sumps could be placed in the context of the whole system leading to more richer water management understanding.

**Strengths:**

Monitoring water level in storage tanks and identifying seepage are important water management requirement in many developing countries, where water is wasted by leaks leading to shortages. The report is well written and informative. It is impressive achievement if this team developed such a compact water monitor.  The methodology and data processing sections are good. The results show interesting findings and a good use of ML to make predictions. The video was explanatory and thank you for sharing the code.

**Weaknesses:**

The report introduction should be been longer with references justifying this research. A description on the need to monitor water level  in storage tanks in region and in other countries with examples of leakage would help set the scene. Regarding the methodology it would be useful to to know how the water system works as a whole in the campus, it is inputs (sumps and pumps) and outputs, and then place the measured nodes in the context of the system. The results are interesting. A little more could be said the behaviour of the IoT devices. The discussion is missing, meaning we cannot fully evaluate the work against it initial objectives.